# Real-Time Road Hazard Information System

**Carlos Pena-Caballero [1], Dongchul Kim [1],\*, Adolfo Gonzalez [1], Osvaldo Castellanos [1], Angel Cantu [1] and Jungseok Ho [2]**

[1] Department of Computer Science, The University of Texas Rio Grande Valley, Edinburgh, TX 78539, USA; carlos.penacaballero01@utrgv.edu (C.P.-C.); adolfo.gonzalez02@utrgv.edu (A.G.); osvaldo.castellanos01@utrgv.edu (O.C.); angel.cantu01@utrgv.edu (A.C.)

[2] Department of Civil Engineering, The University of Texas Rio Grande Valley, Edinburgh, TX 78539, USA; jungseok.ho@utrgv.edu

\* Correspondence: dongchul.kim@utrgv.edu; Tel.: +1-956-665-7923

**Abstract:** Infrastructure is a significant factor in economic growth for systems of government. In order to increase economic productivity, maintaining infrastructure quality is essential. One of the elements of infrastructure is roads. Roads are means which help local and national economies be more productive. Furthermore, road damage such as potholes, debris, or cracks is the cause of many on-road accidents that have cost the lives of many drivers. In this paper, we propose a system that uses Convolutional Neural Networks to detect road degradations without data pre-processing. We utilize the state-of-the-art object detection algorithm, YOLO detector for the system. First, we developed a basic system working on data collecting, pre-processing, and classification. Secondly, we improved the classification performance achieving 97.98% in the overall model testing, and then we utilized pixel-level classification and detection with a method called semantic segmentation. We were able to achieve decent results using this method to detect and classify four different classes (Manhole, Pothole, Blurred Crosswalk, Blurred Street Line). We trained a segmentation model that recognizes the four classes mentioned above and achieved great results with this model allowing the machine to effectively and correctly identify and classify our four classes in an image. Although we obtained excellent accuracy from the detectors, these do not perform particularly well on embedded systems due to their network size. Therefore, we opted for a smaller, less accurate detector that will run in real time on a cheap embedded system, like the Google Coral Dev Board, without needing a powerful and expensive GPU.

**Keywords:** road damage; convolutional neural networks; semantic segmentation

## 1. Introduction

There are many ways in which the manner we maintain our society's quality affects the productivity of locals and national societies. Infrastructures, such as roads, have always been a major factor and reflection in the overall richness of a nation's economy. However, maintaining the quality of roads can be a challenge, and the lack of maintenance can lead to the stagnation of productivity and can even cause on-road accidents. A report from AAA, in 2004 North America [1], showed that over 25,000 crashes per year occurred due to vehicle parts, cargo, or other materials unintentionally discharged from vehicles onto the roadway resulting in approximately 80–90 fatalities; another study by Tefft between 2011–2014 [2] reported over 200,000 police-reported crashes due to road debris, resulting in 39,000 injuries and 500 deaths. The detection of roads that need repair or cleaning is vital in preventing accidents. However, it is challenging for a driver to detect small hazardous objects and damage on the road [3].

We propose a road hazard detection system that will allow the detection of road hazards and allow officials to respond to the hazards quickly. By doing so, we believe that the rate of accidents caused by debris or potholes will decrease considerably. The expectation is that the method will provide officials with accurate detection of hazards that can cause accidents or damage.

There have been many attempts by researchers to solve the road hazard problem. Detection of potholes and cracks on the road has been explored, and algorithms have been developed with good performance [4–7]. Detection of objects, such as planks or tires, has also been explored in the road with excellent performance. However, detection may not always be real-time [8–11]. Different sensor types have been applied to the problem, such as passive cameras, active radar, or laser sensors. While active range sensors provide high accuracy in terms of pointwise distance and velocity measurement, they typically suffer from low resolution and high cost. In contrast, cameras provide very high spatial resolution at a relatively low cost. However, the detection task of small obstacles is a challenging problem from a computer vision perspective, since the considered objects cover tiny image areas and come in all possible shapes and appearances. Another popular method to perform object classification and detection is using semantic segmentation; this algorithm classifies objects at a pixel level, it is used as a masking tool since it effectively can separate an object from others and the background. A study of Pinheiro and Colbbert [12] explored a possibility in which images can be segmented using CNNs; the authors utilized a well-known network called Overfeat, developed by Sermanet et al. [13], and they combined it with a simple CNN to train the segmentation model. It helps them achieve up to 56.25% on the Visual Object Classes (VOC) 2008, 57.01% on the VOC 2009, and 56.12% on the VOC 2010. Chen et al. [14] have developed a more sophisticated state-of-the-art architecture with CNNs, and they have been able to achieve very impressive accuracies on VOC 2012 and Cityscapes [15] datasets, with 71.57% and 71.40%, respectively.

The study of Jo and Ryu [4] described a pothole maintenance system to detect and repair potholes and a method to detect potholes on weak devices (i.e., embedded systems). The system proposed would have a blackbox camera attached to cars that can detect potholes. When the camera detects a pothole, information on the pothole's location and its description would be sent to a database where transportation officials would be able to access and respond. This system would be beneficial due to reduced cost compared to laser methods and enabling a faster response time. The pothole detection method consisted of three steps: pre-processing, choosing candidate features, and cascade detection. The algorithm achieved good accuracy and was robust against false positives. One issue was that, due to the embedded system's hardware, problems with lane detection occurred, but there was no noticeable effect on predictive performance in testing. A study of Eisenbach et al. [5] proposed deep convolutional neural networks that performed very well but are not fit for real-time image capture on vehicles traveling at a medium speed. The network's performance is shown not to be dependent on the size of the network in tests between traditional deep convolutional neural networks and a shallow convolutional neural network described in Zhang et al. [6]. Performance improved on the shallow network when the image block size was increased from $64 \times 64$ to $99 \times 99$. The shallow network is small enough to be capable of running on embedded systems in real-time. In a study of Pauly et al. [7], they tested two convolutional neural networks, one with four layers and the other five, on two datasets, the datasets being pictures of cracks in two different locations. The results show little change in performance between the networks when the same dataset is sampled for training and testing. The five-layer convolutional neural network performed better than the four-layer convolutional neural network when one dataset was used for training, and the other dataset was used for testing. Therefore, deeper neural networks are ideal for generalized detection, but this may incur more significant training time and inference time since as the number of layers increases the time it takes to process an image will increase as well.

In this paper, we will look into different object detection and semantic segmentation algorithms to analyze the best algorithm that will lead to better detection of road damages and better performance able to run independently and at a 'real-time' speed using an embedded system. We will analyze the

methods and data properties that will be used in the paper, evaluate the performance of the detectors used in our experiments, and present a trade-off between the YOLO, the Single Shot MultiBox Detector (SSD) in accuracy and performance. Finally, we present an argument that supports the results we got from the two most recent YOLO [16,17], semantic segmentation [18], and SSD [19] tests.

## 2. Methods

### 2.1. System Overview

It is quite common that older streets in a city tend to wear down and break, causing many different types of damage to the concrete; some of this damage is not dangerous at the beginning, damage such as cracks and blurred lines or markings. However, if left unchecked, they become more dangerous as they can form potholes and leave the streets without proper road markings; these can cause vital problems to vehicles that regularly transit those streets. In this paper, we propose a system, pictured in Figure 1, that starts with an embedded system mounted in a car; this system will be able to recognize, classify, and record all and possible hazards that are in any given location. After the system determines if the road has any hazards, it will report to a web server that will save the image that contains the hazard(s) and the exact location of the hazard using the image coordinates; these images and locations will be saved in the web server, and they will be accessible to the public and authorities for their quick fix. However, this system needs help from authorities and city workers that cruise around the entire city. Since embedded systems tend to be small in size and do not require too much voltage, this system will be able to be mounted on any vehicle without interfering with the driver. All detectors in this paper were trained using a Nvidia GeForce GTX 1080 GPU[20] in a workstation; for testing out the model, we used the Google Coral Dev Board [21]; this embedded system uses a new technology known as the Tensor Processing Unit (TPU), which currently supports a limited set of operations. However, as stated in [22], it should have enough processing power to allow us to run a detection model in real time.

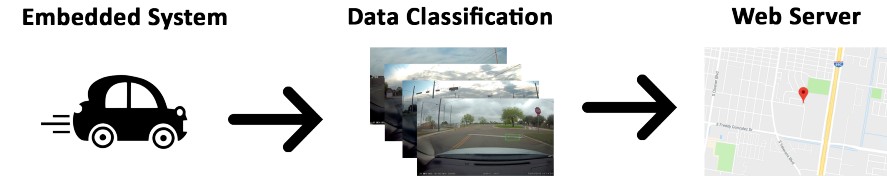

**Figure 1.** Overview of the proposed system; the embedded system mounted on a volunteers car will capture images while driving normally; then, the system will process the images in real time and send data to a remote server only if the image contains at least one of our proposed classes; the server will then save the image and location of said class.

### 2.2. Object Detection

#### 2.2.1. Convolutional Neural Network

Artificial Neural Networks (ANNs) are computational networks that attempt to simulate the decision process in a biological neural network [23]. ANNs have multiple layers (at least three) as shown in Figure 2, each neuron is fully connected to each other neuron in the next layer, and they can be viewed as weighted directed graphs in which artificial neurons are nodes and directed edges (with weights) are connections between neuron output and neuron input [24]; these connections represent a series of calculations that determines the nature of the input and outputs either a binary (one output neuron) or a multi-class (a vector of neurons) value, the connections will change weights over time as the network trains, if a certain neural path does not lead to a true positive prediction, then the weight of the connections in the path will be lowered so the network is less likely to predict the same output, thus giving the ability to learn and recognize underlying patterns rather than following a set of rules

specified by human experts. This method of learning will continue until the loss (the prediction errors) is very small or a certain number of epochs are completed. Many experiments have shown that ANNs are particularly good with natural data such as speech [25,26], language [27,28], or vision [29,30].

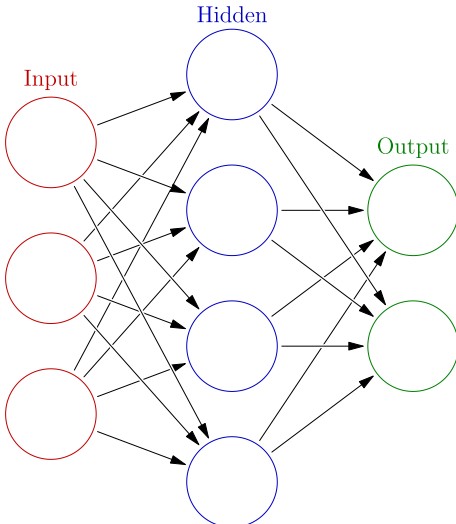

**Figure 2.** Example of ANNs, the number of input neurons determines the number of features considered in the network, every neuron is connected by a weight and will determine the output layer's prediction.

Image classification is a classic machine learning problem [29–31]. It is quite simple for a human observer to recognize or identify an object in an image, i.e., handwritten digits, road damages, etc., but for a computer this is a daunting task, there are many models for image classification which fit the training points but do not generalize well to unseen images, i.e., Template Matching [32], State Vector Machines [33], or Harr Cascade Classifiers [34]. Convolutional Neural Networks (CNNs), on the other hand, is very effective in image recognition and similar tasks, such as object detection or tracking. CNNs used for image classification take a three-dimensional input image (height, width, and channel); height and width represent the number of pixels in the image, and the channel will define the depth of the image, either grayscale (one channel) or RGB (three channels). The input then sequentially goes through a series of processing steps, which could be a convolutional layer, a pooling layer, a fully connected layer, etc. [35]. Convolutional layers utilize a kernel to scan the pixel data of the image and generate a smaller set of features that will be passed to the next layer; pooling layers will perform a downsampling operation along the spatial dimensions (height, width); and fully connected layers will compute class scores using the features from the previous layer.

The main difference between an image classification and image detection algorithm is that, in classification, we determine the class of the image thus a single output class, but, in detection, we want to draw a bounding box around the object in the image, and this requires the network to return a location on the screen the size of the bounding box and the class of the object; there can be many objects in a single image; thus, the output layer will have a variable size so regular CNNs like the ones described above do not work for this purpose. In the study of Girshick et al. [36], the authors proposed an object detection algorithm called R-CNN; this algorithm handled detection by extracting 2000 regions from the input image and recursively combining similar regions to create what the authors called "region proposals"; then, each region's proposal would be resized and fed into a CNN which would then determine the class of that region. This algorithm was then improved in Fast R-CNN by Girshick [37].

### 2.2.2. YOLO Detector

The detection algorithms introduced in the previous section have one drawback, and that is the pre-processing time it takes to generate the region proposals, and each resulting proposal would have to be classified individually; thus, the network never looks at the entire image. The YOLO (You only look once) Detector [38] does not waste time generating the region proposals, but rather it will input an image split in SxS default regions, each region in the input would be processed at the same time and each will output with a list of possible objects and a level of confidence, the regions that reach above a certain threshold of confidence would be selected to locate the object within the image. To begin working on this project, we opted to use a fast and accurate detector called YOLO 9000 [16]; we chose this detector because it is the fastest detector today peeking at more than 100 Frames Per Second (FPS) on a medium quality video. The network's architecture for this detector can be visualized in Table 1. We will also look into a new and experimental version of YOLO, which introduces significantly more layers in its architecture, and we hope it will increase the accuracy of the hard classes that we have in our experiment.

**Table 1.** YOLO 9000 Network's Architecture Darknet-19 [16].

| Type | Filters | Size/Stride | Output |
| --- | --- | --- | --- |
| Convolutional | 32 | $3 \times 3$ | $224 \times 224$ |
| Maxpool | | $2 \times 2/2$ | $112 \times 112$ |
| Convolutional | 64 | $3 \times 3$ | $112 \times 112$ |
| Maxpool | | $2 \times 2/2$ | $56 \times 56$ |
| Convolutional | 128 | $3 \times 3$ | $56 \times 56$ |
| Convolutional | 64 | $1 \times 1$ | $56 \times 56$ |
| Convolutional | 128 | $3 \times 3$ | $56 \times 56$ |
| Maxpool | | $2 \times 2/2$ | $28 \times 28$ |
| Convolutional | 256 | $3 \times 3$ | $28 \times 28$ |
| Convolutional | 128 | $1 \times 1$ | $28 \times 28$ |
| Convolutional | 256 | $3 \times 3$ | $28 \times 28$ |
| Maxpool | | $2 \times 2/2$ | $14 \times 14$ |
| Convolutional | 512 | $3 \times 3$ | $14 \times 14$ |
| Convolutional | 256 | $1 \times 1$ | $14 \times 14$ |
| Convolutional | 512 | $3 \times 3$ | $14 \times 14$ |
| Convolutional | 256 | $1 \times 1$ | $14 \times 14$ |
| Convolutional | 512 | $3 \times 3$ | $14 \times 14$ |
| Maxpool | | $2 \times 2/2$ | $7 \times 7$ |
| Convolutional | 1024 | $3 \times 3$ | $7 \times 7$ |
| Convolutional | 512 | $1 \times 1$ | $7 \times 7$ |
| Convolutional | 1024 | $3 \times 3$ | $7 \times 7$ |
| Convolutional | 512 | $1 \times 1$ | $7 \times 7$ |
| Convolutional | 1024 | $3 \times 3$ | $7 \times 7$ |
| - | - | - | - |
| Convolutional | 1000 | $1 \times 1$ | $7 \times 7$ |
| Avgpool | | Global | 1000 |
| Softmax | | | |

## 3. Results

### 3.1. Data

As seen in [39], the data can be easily acquired by using a smartphone mounted on the dashboard of a vehicle. We recorded various streets from our local cities in Texas, McAllen, Mission, and Edinburg, and then used a Python script to extract pictures from the recorded videos; see Figure 3 to observe the streets we recorded on each city. The green mark represents data that have been annotated to be used in our training and testing samples, but yellow is still raw data left to annotate. After we

extracted our data, we manually select specific pictures where a particular road damage type can be seen. We help generalize our data by having recorded in various kinds of weather. Having acquired our data, we then use a Python program, which allows us to manually label each incident of damage spotted in our data. This way, we can be sure that every recorded pavement damage will be recorded and learned by our network. With our methods, there is no pre-processing of our data; the use of a mobile device to acquire our data makes our methods more versatile and flexible regarding the ease of acquiring data and utilizing it. We have collected more than 4000 images of various roads using an iPhone 7 with 4K quality[40], from which we were able to annotate 1700 resulting in the distribution of labels shown in Figure 4.

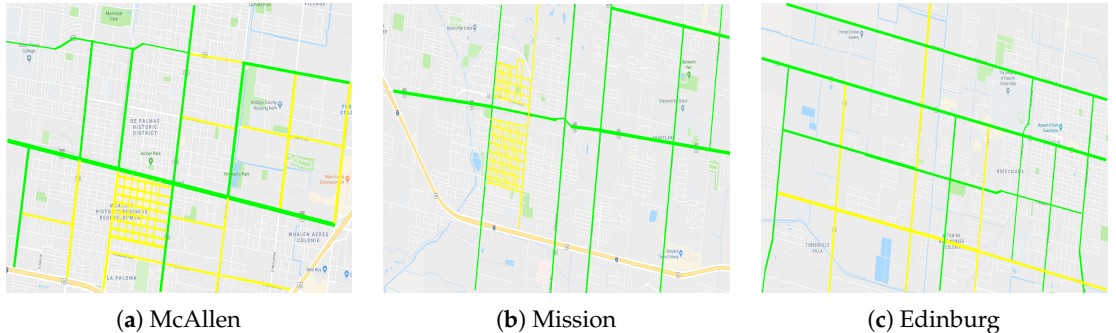

| (**a**) McAllen | (**b**) Mission | (**c**) Edinburg |
|---|---|---|

**Figure 3.** Marked spots in the map represent the streets we recorded to obtain our train and test data, green marks are annotated and yellow are pending.

### 3.2. Evaluation

The object detector is evaluated by using a method called Intersection-Over-Union (IOU) from which the predicted boxes from our trained detector are compared to our ground truth boxes from our labeled set. According to Everingham et al. [41], in order for a predicted box to be counted as a True Positive, it has to be at least 50% of the box inside the ground truth; using this, we can calculate the Average Precision (AP) of each class and then determine the mean AP (mAP) of the model. Utilizing Equation (1), we can calculate the IOU of each object's predicted bounding box.

$$a_0 = \frac{area(B_p \bigcap B_{gt})}{area(B_p \bigcup B_{gt})} \tag{1}$$

where $a_0$ denotes the percentage of area shared between the two bounding boxes, $B_p$ and $B_{gt}$ are the area of the predicted bounding box and ground truth bounding box, respectively. This equation will calculate the intersection of $B_p$ and $B_{gt}$ and divide it over the union of $B_p$ or $B_{gt}$.

The holdout validation strategy is considered to be a reliable way to estimate the accuracy of a prediction model [42–47]; this strategy will randomly divide our available dataset into two subsets: train and test sets; the training set will be used to fit our model and learn the features of the object classes; then, the test set will present new, unseen data to the model. The most common train and test split for holdout are 2/3 training data and 1/3 testing data, but, since our dataset is relatively small, we opted to split our dataset as follows: we randomly assigned 90% (1̃530 images) of the dataset as training data and 10% (1̃70 images) as test data, while ensuring every class had sample images in both sets; we then trained the model and use the test set to evaluate its mAP; we repeated this process 10 times and obtained the mAP over all 10 tests.

### 3.3. YOLOv2

Our trained detector included seven different classes, and the resulting IOU of our single test is 69.58% mAP in which we obtained 151 of true positive correctly predicted boxes over 66 of false-negative incorrectly predicted boxes. We can observe some correct predictions and difficult

cases in Figures 5 and 6 as well as the more challenging images that were not predicted on our test of the detector in Figure 7. Table 2 shows resulting mAP for each class using the distribution shown in Figure 4.

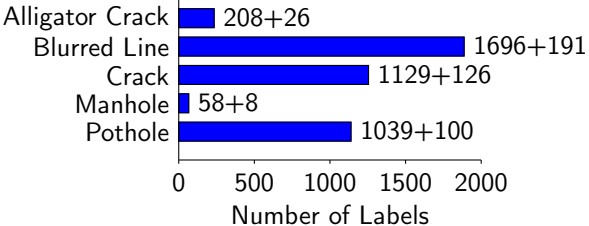

**Figure 4.** Distribution of train + test labels used in our detector's training and evaluation (test images were not included at the time of training).

**Table 2.** Mean Average Precision (mAP) per class for both detectors, and they were tested using the same number and distribution of training and testing images.

|  | mAP | |
| --- | --- | --- |
| **Classes** | **YOLOv2** | **YOLOv3** |
| Longitudinal Crack | 59.62% | 99.71% |
| Lateral Crack | 42.86% | 100.0% |
| Manhole | 62.50% | 100.0% |
| Alligator Crack | 18.75% | 98.54% |
| Pothole | 90.00% | 98.82% |
| Blurred Streed Line | 91.67% | 98.33% |
| Blurred Crosswalk | 52.63% | 90.48% |

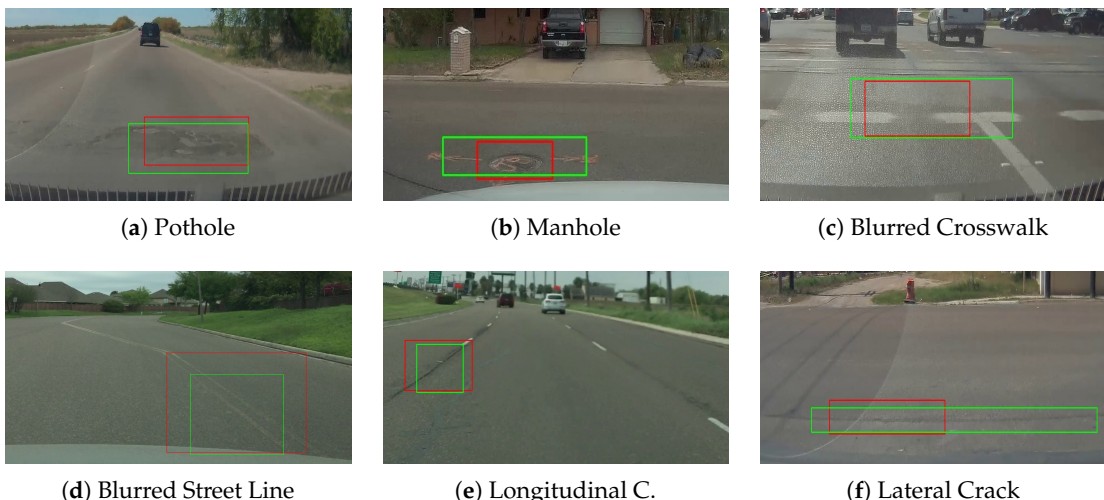

(**a**) Pothole　　　　　　(**b**) Manhole　　　　　　(**c**) Blurred Crosswalk

(**d**) Blurred Street Line　　　　(**e**) Longitudinal C.　　　　(**f**) Lateral Crack

**Figure 5.** Examples of true positive predicted bounding boxes, the box in red is the ground truth, and the box in green represents the predicted bounding box.

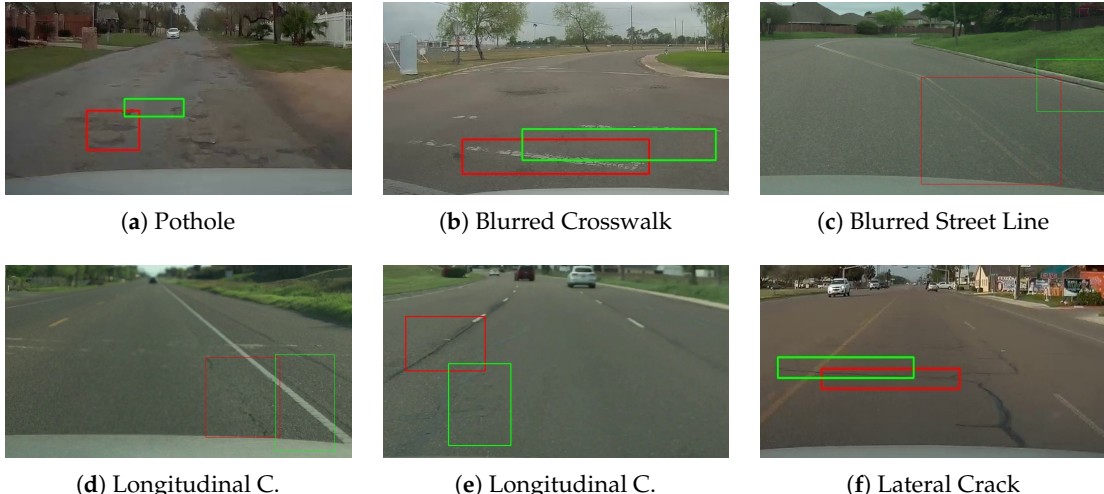

(**a**) Pothole       (**b**) Blurred Crosswalk       (**c**) Blurred Street Line

(**d**) Longitudinal C.       (**e**) Longitudinal C.       (**f**) Lateral Crack

**Figure 6.** Examples of false positive predicted bounding boxes; notice that, in order to be considered true positive, the IOU has to be equal to or greater than 50%.

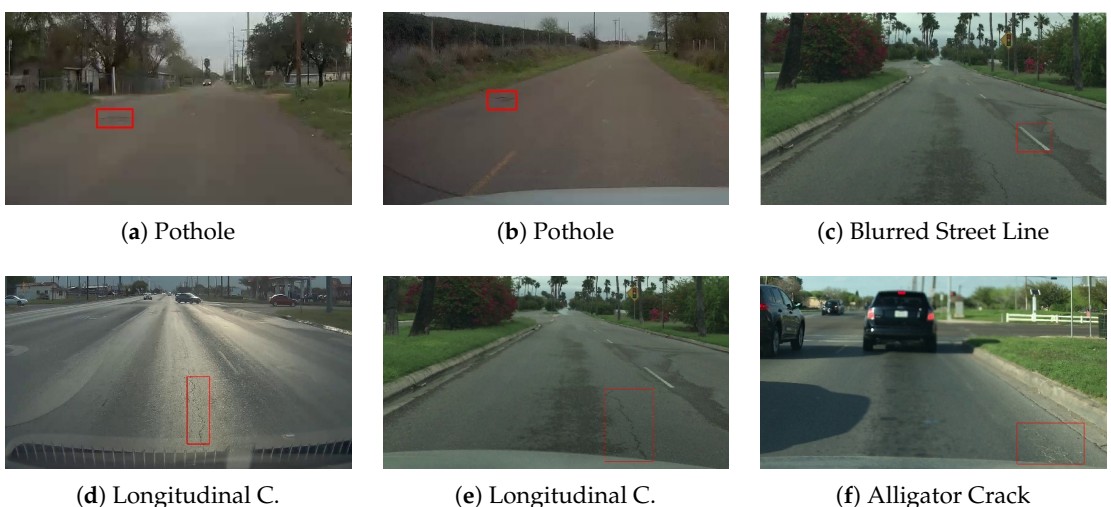

(**a**) Pothole       (**b**) Pothole       (**c**) Blurred Street Line

(**d**) Longitudinal C.       (**e**) Longitudinal C.       (**f**) Alligator Crack

**Figure 7.** Examples of difficult false negatives, images that the detector was not able to predict.

### 3.4. YOLOv3

YOLO's recent paper provided a new architecture for the detection algorithm [17], it can be visualized in Table 3; the process is still experimental, but it appears to be performing surprisingly faster than the previous version of YOLO; the main improvement of the detector is in the number of their CNN layers it contains; this new model includes 53 layers of CNNs almost three times the architecture of Table 1. During this stage of the experiment, we kept the same training and testing images from Figure 4, the results of this new architecture were incredibly surprising, the network performed an overall mAP of 97.98%, and we can observe all the classes precision for this detector in Table 2.

**Table 3.** YOLO version 3 Network's Architecture Darknet-53 [17].

|  | Type | Filters | Size/Stride | Output |
|---|---|---|---|---|
|  | Convolutional | 32 | $3 \times 3$ | $256 \times 256$ |
|  | Convolutional | 64 | $3 \times 3$ | $128 \times 128$ |
|  | Convolutional | 32 | $1 \times 1$ |  |
| $1\times$ | Convolutional | 64 | $3 \times 3$ |  |
|  | Residual |  |  | $128 \times 128$ |
|  | Convolutional | 128 | $3 \times 3/2$ | $64 \times 64$ |
|  | Convolutional | 64 | $1 \times 1$ |  |
| $2\times$ | Convolutional | 128 | $3 \times 3$ |  |
|  | Residual |  |  | $64 \times 64$ |
|  | Convolutional | 256 | $3 \times 3/2$ | $32 \times 32$ |
|  | Convolutional | 128 | $1 \times 1$ |  |
| $8\times$ | Convolutional | 256 | $3 \times 3$ |  |
|  | Residual |  |  | $32 \times 32$ |
|  | Convolutional | 512 | $3 \times 3/2$ | $16 \times 16$ |
|  | Convolutional | 256 | $1 \times 1$ |  |
| $8\times$ | Convolutional | 512 | $3 \times 3$ |  |
|  | Residual |  |  | $16 \times 16$ |
|  | Convolutional | 1024 | $3 \times 3/2$ | $8 \times 8$ |
|  | Convolutional | 512 | $1 \times 1$ |  |
| $4\times$ | Convolutional | 1024 | $3 \times 3$ |  |
|  | Residual |  |  | $8 \times 8$ |
|  | Avgpool |  | Global |  |
|  | Connected |  | 1000 |  |
|  | Softmax |  |  |  |

### 3.5. Semantic Segmentation

In object detection, the network must analyze an image and return the object's class as well as the position and size within the image; these objects are defined by a label and a bounding box in the dataset, these bounding boxes may overlap the same object's class, thus making the inference of many objects of the same class in a given region difficult. Because drawing boxes around objects does not give an accurate idea of the object's shape, the next natural step in the progression from coarse to fine inference is to make a prediction at every pixel [48]. In Semantic Segmentation, an image input will be processed and downsampled during the first half of the network; then, it will upsample the extracted features so the network outputs an image of the same size as the input, except, it will include only the segments for the objects found, perhaps the easiest way to visualize this algorithm is to look at Figure 8. Some segmentation samples can be seen in Figures 9 and 10.

This new approach required that we used brand new annotations since the typical Pascal VOC annotations are only bounding boxes around the objects, and we used a very useful online tool called LabelBox (https://www.labelbox.com) to help us annotate our current dataset; for this experiment, we trained on four different classes (Manhole, Pothole, Blurred Crosswalk, Blurred Street Line) with the architecture of FC-DenseNet103 [18]. A few sample validation images from the training process of the model can be seen in Figure 9; for the most part, the segmentation process went near-optimal, but the model had trouble isolating and recognizing only the damaged part of the Street Lines; as you can see in Figure 10c, the model was segmenting the street lines completely. Using these results, we improved our labeling process by labeling not only the damaged parts but the entire object. Our final segmentation model had all four classes included in their training and test samples. In Figure 11, we can see some of the results obtained after the training process; the predicted image in this model detected hazards that were missed by during the annotation process and, in the case of Figure 11c, the predicted image is even more accurate than the original ground truth.

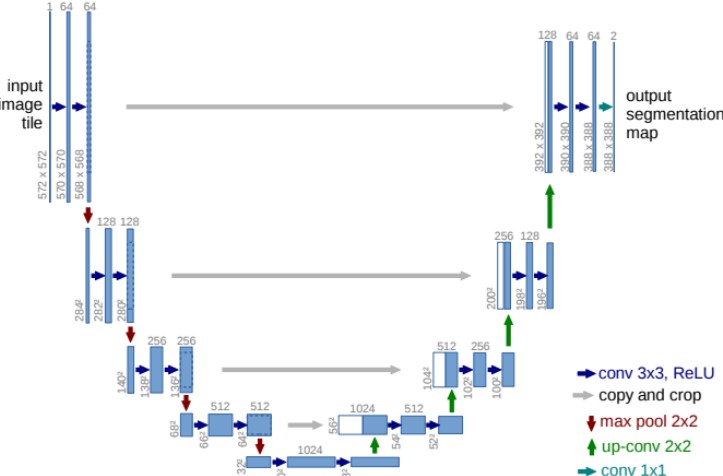

**Figure 8.** U-net architecture (example for 32 × 32 pixels in the lowest resolution). Each blue box corresponds to a multi-channel feature map. The number of channels is denoted on top of the box. The *x–y*-size is provided at the lower left edge of the box. White boxes represent copied feature maps. The arrows denote the different operations [49].

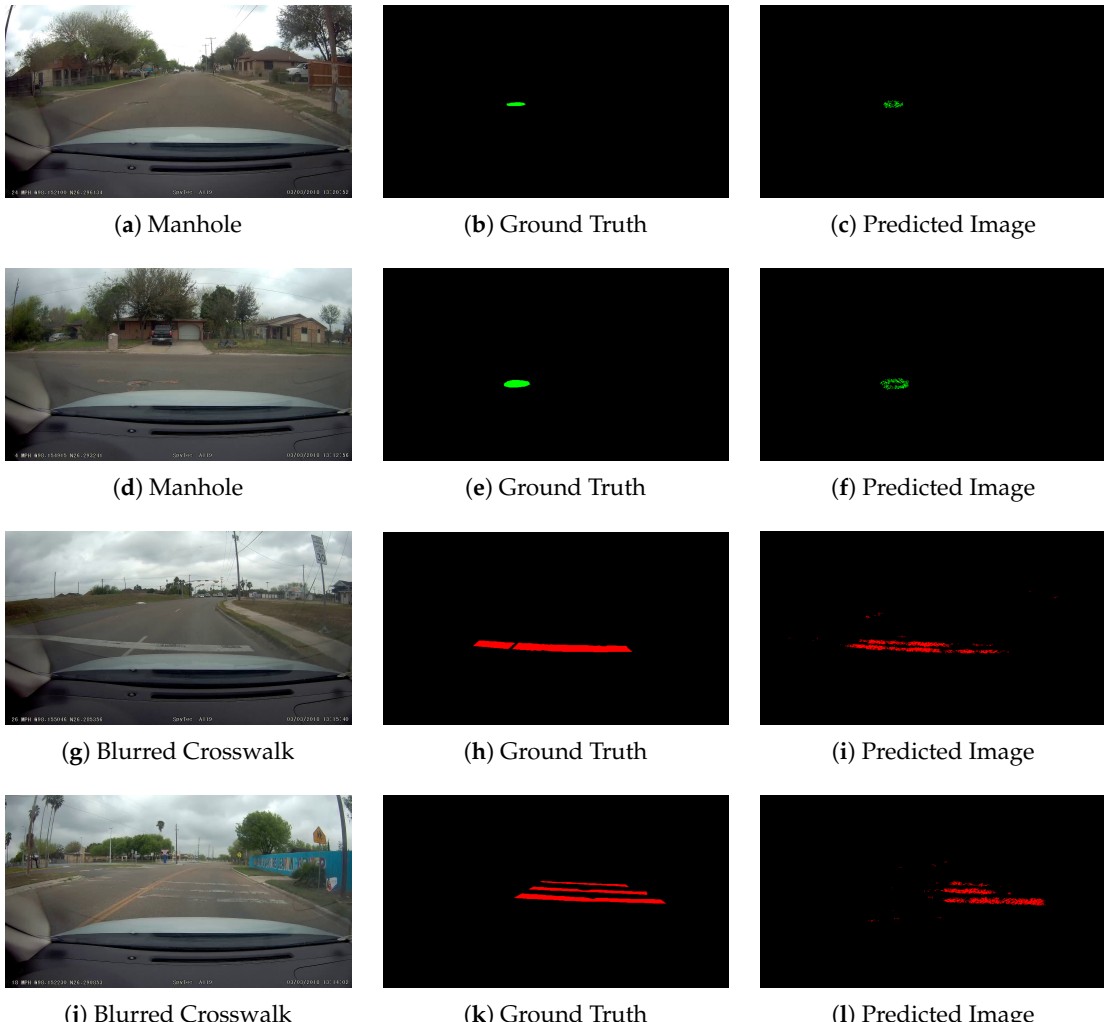

**Figure 9.** Semantic segmentation examples on FC-DenseNet 130 [18].

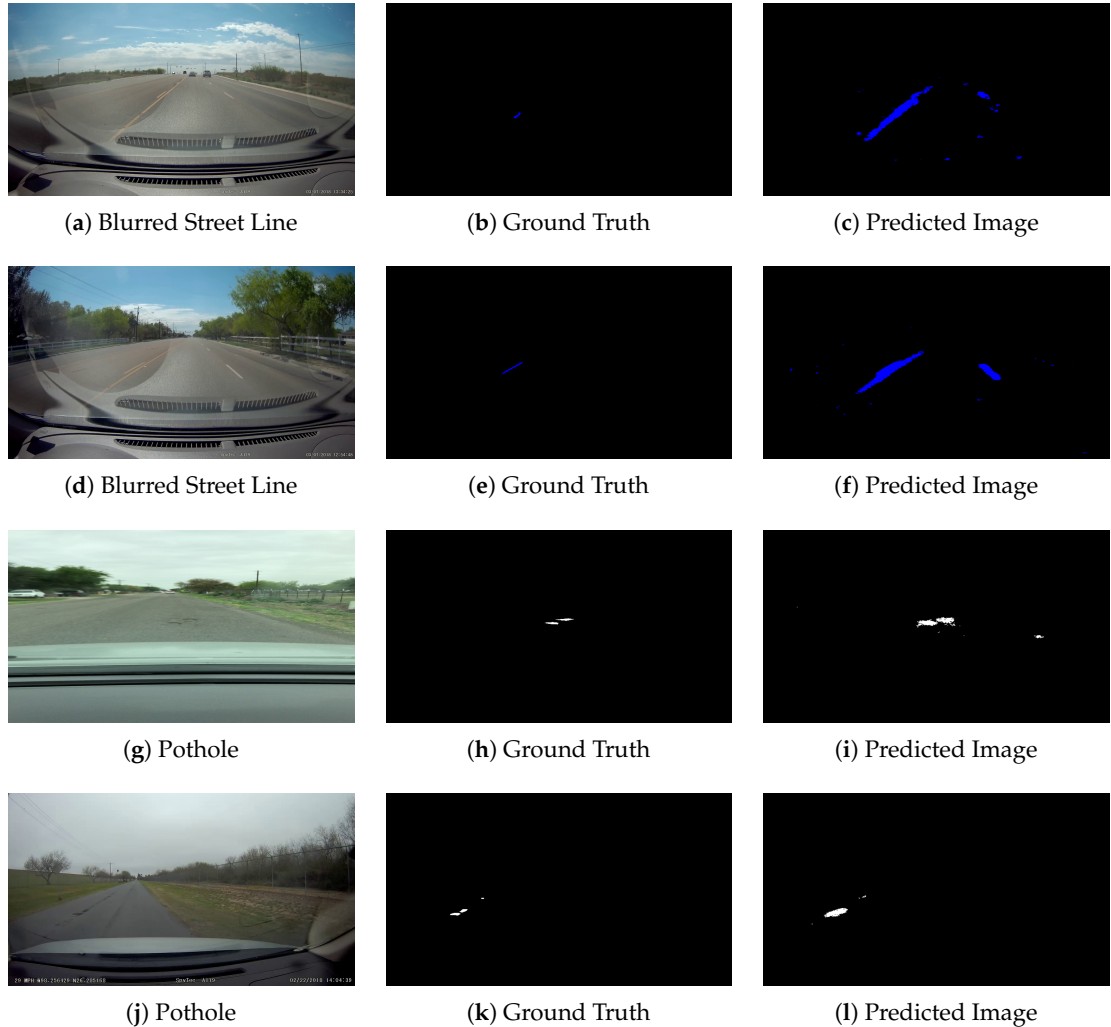

**Figure 10.** Semantic segmentation examples on FC-DenseNet 130 [18].

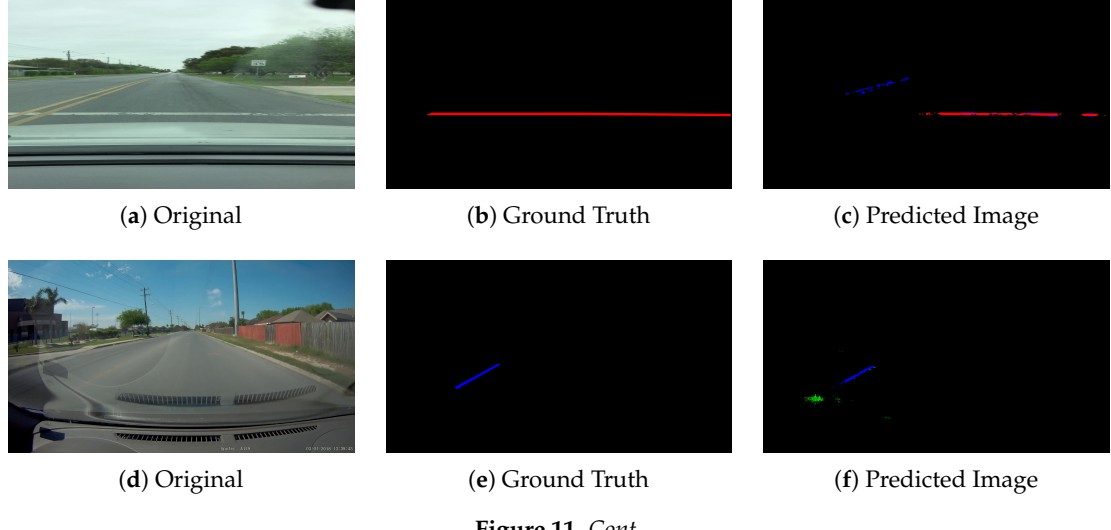

**Figure 11.** *Cont.*

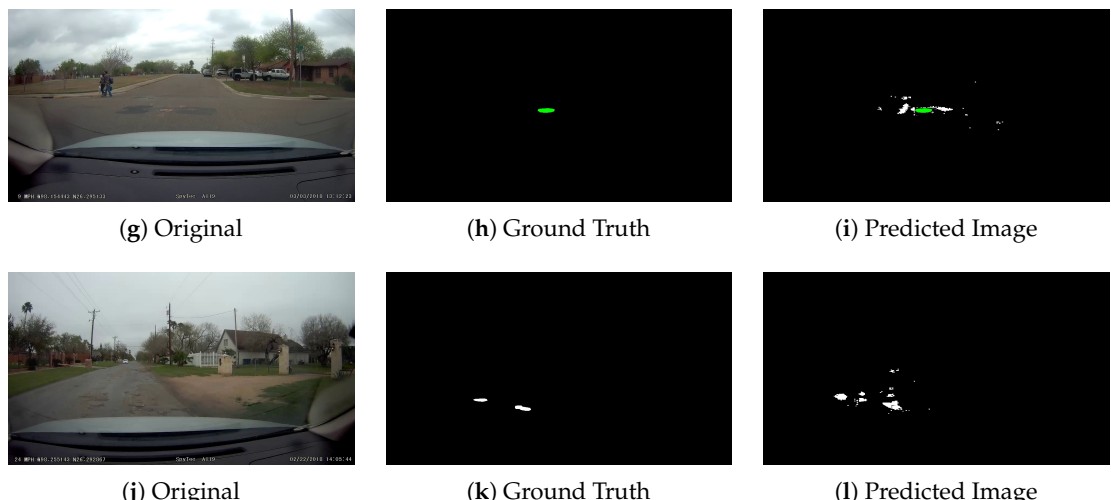

(**g**) Original        (**h**) Ground Truth        (**i**) Predicted Image

(**j**) Original        (**k**) Ground Truth        (**l**) Predicted Image

**Figure 11.** Semantic segmentation examples with four classes using FC-DenseNet 130 [18].

### 3.6. Google Coral Dev Board

Our chosen embedded system is called Google Coral Dev Board [21], its retail cost starts at 129.99 USD, this board features a Quad-core CPU NXP i.MX 8M SoC, Integrated GC7000 Lite Graphics, 1GB LPDDR4 of RAM, 8 GB eMMC of Flash Memory, and an ML accelerator called the Google Edge Tensor Processing Unit (TPU); this board requires only a 5V DC USB Type-C to power and its dimensions are 88 mm × 60 mm × 24 mm. Compared to the widely known Raspberry Pi 3B+, this board is more expensive but way faster and provides a better experience, and even though there other powerful embedded systems currently in the market such as the Jetson nano (retailing at 99.99 USD), the Google Coral provides a more native conversion of Tensorflow models to TPU accelerated models. The TPU is an AI accelerator highly-optimized for large batches and CNNs. However, only a small limited set of operations are available to be used in the TPU; this limits the size of the network we can use in our board, and this means that training a model to run on this TPU must use a quantization-aware training. Therefore, we cannot use any regular network architecture, such as YOLO; instead, we must use a Tensorflow lite model, which is considerably smaller and has a reduced accuracy so the model inference can be done in real time.

### 3.7. Single Shot Multibox Detector (SSD)

The SSD approach is based on a feed-forward convolutional network that produces a fixed-size collection of bounding boxes and scores for the presence of object class instances in those boxes, followed by a non-maximum suppression step to produce the final detections. We will use a unique SSD architecture known as SSD 300 [19]; this means that the images in our dataset will be scaled down to 300 by 300 pixels. MobileNet is a convolutional network architecture design to be used on systems that lack processing power, embedded systems, and mobile devices. MobileNets are based on a streamlined architecture that uses depth-wise separable convolutions to build lightweight deep neural networks [50]. Because this network is geared towards mobile devices, its mAP tends to be worse than most of the other CNN architectures. MobileNetV2 improved the overall accuracy and speed of its predecessor, which is why we chose to use this version; we can see the architecture for this network in Table 4. During the early testing of the SSD 300 MobileNetV2 architecture, we found that the model struggled to differentiate between a lateral and a longitudinal crack, as well as the blurred crosswalk and street lines; these classes are especially challenging since they look very similar. Therefore, we decided to reduce our original seven classes to only five, Pothole, Manhole, Blurred Line, Crack, and Alligator Crack. Table 5 shows the mAP for each class after the training completes; by far, the easiest classes to detect are the Blurred Lines and Manhole covers; this is because these classes tend

to be very uniform; thus, it does not take too long for the model to learn its patterns. However, on the other hand, we can see that cracks are the most difficult to detect; this is because the class has too many different variations on how a crack would appear in the street, the area of many of these labels is too small, and many have an inferior resolution when it got scaled down. On Figures 12 and 13, we can see a few examples of true positive and false positive predictions of the detector. Although this model did not produce a lot of false positive predictions, it is worth mentioning that, similar to YOLO, the SSD model predicted labels that were missed during the annotation process. This known property lets us argue that our model can start detecting hazards with its current accuracy of 41.83%, and produce predictions that may be added to our model's dataset once a human observer determines their accuracy. With this process, we are able to tailor our model for a specific location.

**Table 4.** MobileNetV2 : Each line describes a sequence of 1 or more identical (modulo stride) layers, repeated *n* times. All layers in the same sequence have the same number c of output channels. The first layer of each sequence has a stride s and all others use stride 1. All spatial convolutions use $3 \times 3$ kernels [51].

| Input | Operator | t | c | n | s |
|---|---|---|---|---|---|
| $224^2 \times 3$ | conv2d | - | 32 | 1 | 2 |
| $112^2 \times 32$ | bottleneck | 1 | 16 | 1 | 1 |
| $112^2 \times 16$ | bottleneck | 6 | 24 | 2 | 2 |
| $56^2 \times 24$ | bottleneck | 6 | 32 | 3 | 2 |
| $28^2 \times 32$ | bottleneck | 6 | 64 | 4 | 2 |
| $14^2 \times 64$ | bottleneck | 6 | 96 | 3 | 1 |
| $14^2 \times 96$ | bottleneck | 6 | 160 | 3 | 2 |
| $7^2 \times 160$ | bottleneck | 6 | 320 | 1 | 1 |
| $7^2 \times 320$ | conv2d $1 \times 1$ | - | 1280 | 1 | 1 |
| $7^2 \times 1280$ | avgpool $7 \times 7$ | - | - | 1 | - |
| $1 \times 1 \times 1280$ | conv2d $1 \times 1$ | - | k | | - |

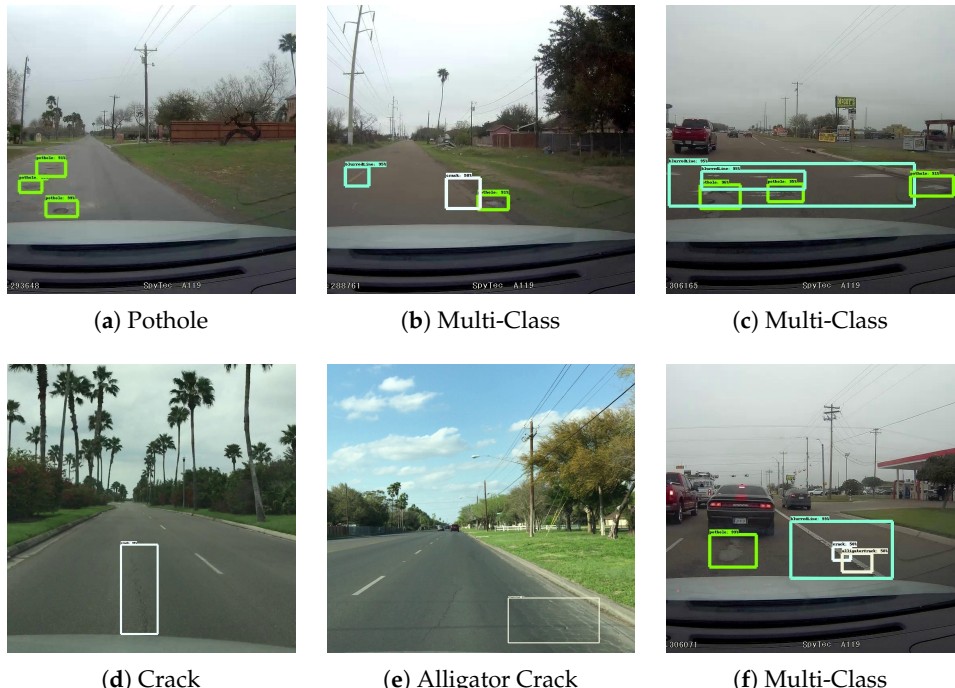

(**a**) Pothole  (**b**) Multi-Class  (**c**) Multi-Class

(**d**) Crack  (**e**) Alligator Crack  (**f**) Multi-Class

**Figure 12.** Examples of true positive predicted bounding boxes for the SSD300 MobileNetV2 [19,51] trained model.

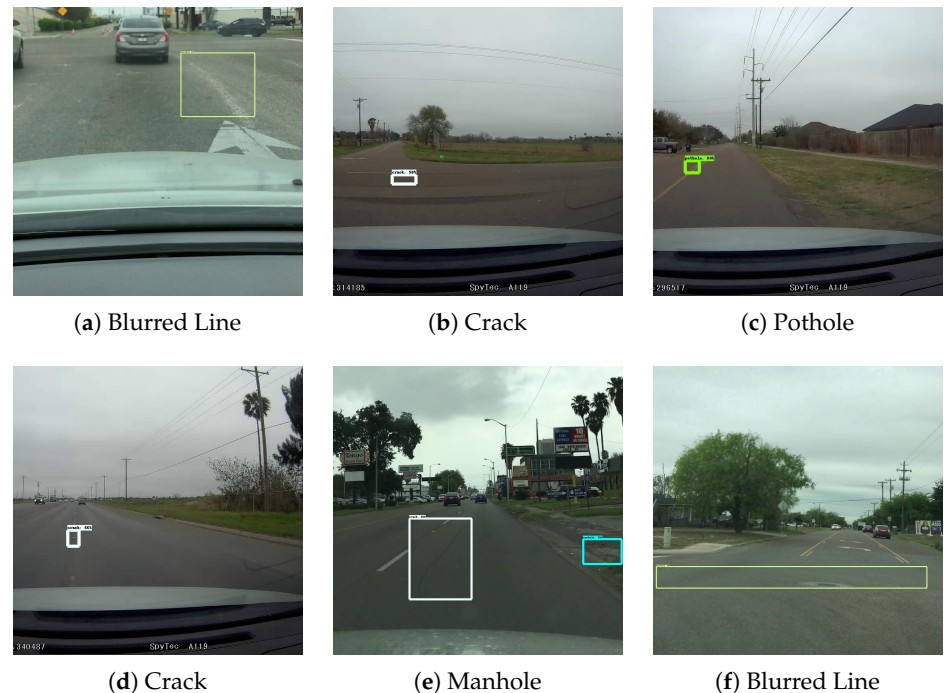

(**a**) Blurred Line        (**b**) Crack        (**c**) Pothole

(**d**) Crack        (**e**) Manhole        (**f**) Blurred Line

**Figure 13.** Examples of false positive predicted bounding boxes for the SSD300 MobileNetV2 [19,51] trained model.

**Table 5.** SSD300 MobileNetV2 mean Average Precision (mAP) per class.

| Classes | mAP |
|---|---|
| Pothole | 45.10% |
| Manhole | 47.14% |
| Blurred Line | 47.78% |
| Crack | 30.14% |
| Alligator Crack | 39.00% |

## 4. Discussion and Conclusions

Overall, the detector is performing above average, considering the small dataset on which we trained the detector. We consider these results acceptable for the size of our dataset since conventional detectors training of the Pascal VOC dataset 2007 + 2012 has a total of 9963 + 11,530 (21,493) images which contain 24,640 + 27,450 (52,090) labels divided among 20 different object classes. Table 6 below shows the mAP of selected detectors on this dataset. However, the true lack of power in these detectors shows when tested on a much larger dataset like MS COCO, which contains more than 200,000 images and over 500,000 object instances among 80 object classes; in Table 7, we can see that the detectors struggle to get their mAP above 50%. During the results of YOLOv2 versus the YOLOv3, we observed a significant increase in our detection accuracy. Although this came with a trade-off with the detection speed, YOLOv3 runs at almost 1/3 the original velocity of what YOLOv2 was achieving. Thus, we still need to weight off both spectrums of the detection since we need our detection to happen in real-time, but a higher accuracy would result in less false negative detections, thus yielding a better system. Image segmentation can provide us pixel-perfect hazard predictions which could be used to further classify the hazards, but it would come with a high computational cost that will result in a less than optimal "real-time" system. However, the SSD detector was not able to perform as well as those previous detectors, but it was the only one that achieved "real-time" processing in our embedded system, Google Coral. This model's inference time per image takes about 30 ms, which allows our

model to process up to 30 FPS; a car using our system could drive at 30 mph and detect hazards every 15 inches. Using this system, a city could deploy several Google Corals on vehicles that regularly transit the streets, the low accuracy shown in the results can be further improved if we can add more data to the dataset, but this can be easily achieved when the system is deployed and starts detecting other hazards on the road.

**Table 6.** Comparison of accuracy and speed between detectors on the Pascal VOC dataset.

| System | Pascal VOC | mAP | FPS |
| --- | --- | --- | --- |
| Fast R-CNN [37] | 2007 + 2012 | 70.0 | 0.5 |
| Faster R-CNN VGG-16 [37] | 2007 + 2012 | 73.2 | 7 |
| YOLO [38] | 2007 + 2012 | 63.4 | 45 |
| YOLOv2 544 × 544 [16] | 2007 + 2012 | 78.6 | 40 |
| SSD300 [19] | 2007 + 2012 | 74.3 | 46 |
| SSD512 [19] | 2007 + 2012 | 76.8 | 19 |

**Table 7.** Comparison of accuracy between detectors on the COCO dataset.

| System | COCO 2015 | mAP |
| --- | --- | --- |
| Fast R-CNN [37] | train | 35.9 |
| Faster R-CNN VGG-16 [37] | trainval | 42.7 |
| SSD300 [19] | trainval35k | 45.3 |
| SSD512 [19] | trainval35k | 41.2 |
| YOLOv2 544 × 544 [16] | trainval35k | 46.5 |
| YOLOv3 608 × 608 [17] | trainval35k | **57.9** |

**Author Contributions:** Conceptualization, methodology, and writing–original draft preparation, C.P.-C.; software and data curation, A.G., O.C., and A.C.; review and editing, J.H.; supervision, D.K. All authors have read and agreed to the published version of the manuscript.

**Funding:** This work was supported by Dwight David Eisenhower Transportation Fellowship Program funded by Federal Highway Administration, U.S. Department of Transportation.

**Acknowledgments:** The authors would like to thank NVIDIA for the GPU grant.

**Conflicts of Interest:** The authors declare no conflict of interest.

## Abbreviations

The following abbreviations are used in this manuscript:

| | |
| --- | --- |
| MDPI | Multidisciplinary Digital Publishing Institute |
| YOLO | You Only Look Once |
| SSD | Single Shot Multibox Detector |
| TPU | Tensor Processing Unit |
| CPU | Central Processing Unit |
| GPU | Graphics Processing Unit |
| RAM | Random Access Memory |
| VOC | Visual Object Classes |
| ANN | Artificial Neural Network |
| CNN | Convolutional Neural Network |
| FPS | Frames Per Second |
| IOU | Intersection Over Union |
| mAP | mean Average Precision |

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
