# Peer review of "Real-Time Road Hazard Information System"

_infrastructures, doi:10.3390/infrastructures5090075_

Round 1

Reviewer 1 Report

Some matters from my previous review were addressed, but some others were not.

These issues require further work:
- The authors describe the foundations of vanilla classification CNNs, but they use Region-CNNs and Pixel-wise CNNs. The section 2.2.2 includes a brief description of YOLO that do not exlain how it works. Explanations about pixel wise classification is completely missing.
- The training/testing/validation protocol is not clear. There are many missing details about the dataset too. How many samples are in each training/test/validation splits? It is important to use a proper validation protocol in order to avoid underfitting/overfitting

These matters should be addressed before considering this paper for publication.

Author Response

Attached is a word document with my comments for the reviewer.

Reviewer 2 Report

General comments

The authors are using AI techniques to detect road degradations. This manuscript could be a nice and useful contribution to its scientific field. However, some major issues must be resolved in order the manuscript be suitable for publication.

Specific comments

Line 5 : the characterization “unfortunate” is not necessary, please remove it

Line 25: for which area is this study referring to?

Line 26-28: “or other….2011” please site the associated reference

Line 47: “ on [11]” please replace/revise with “in a study of -authors names- [11]’’

Line 63: [15]: the same as above

Line 69: [6]: the same as above

Line 75: please articulate about the training time and runtime

Line 104: the correct abbreviation is ANNs, please replace. Why are you using the term “deep”? what about shallow neural networks? The way you define ANNs as a deep learning technique is not correct. ANNs are characterized as an   information   processing system which contains a   large   number   of   highly interconnected processing neurons. Please revise this and use some associated references for your definition.

Line 105: please use some studies/references where ANNs are used for regression and classification

Line 105-106: “although….problem” please be more specific about these variations and the associated problem. Also, mention 1-2 related case studies

Line 116: “just like ANNs” ? CNNs are ANNs, please revise

Lines 116-120:  “use ….input)” these lines are written in a fuzzy/ unclear way. Please eliminate or revise substantially. In case of revision, citing the associated literature would be helpful.

Line 118-119: the table’s legend should be above from Table 1.

Also, placing tables site by site is not recommended. Same for the rest of tables, through out the manuscript.

Lines 120-122: “Then….classification” : adjust the weights until when? Please be more specific/articulate about the weights adjustment process.

Lines 122-126: “For example….manhole” I consider these lines unnecessary. Please eliminate them

Line 138: Figure number is missing

Line 152: same as line 47 comment

Line 155: “using 1” please add the word equation.  Also regarding equation 1, mention for each of its terms what it stands for.

Line 160: “ and 4” the word figure is missing

Line 168: what do you mean with the abbreviation “mAP” ? please check it for the rest of the text

Line 169-170: Figure 6: please write the x-axis legend

Line 170- 185: Regarding Semantic Segmentation part, I recommend the addition of a flow chart explaining clearly the methods/process you apply. Furthermore, this whole part needs revising in order to clarify+ explain analytically the methods and steps you apply.

Line 181: Please explain further how you improved your labelling process

Line 184: it shouldn’t be Figure 9.1, please correct it

Line 204: please remove [18]

Line 246: please remove *

Author Response

(The authors gave the same response as above.)

Round 2

Reviewer 1 Report

The authors used a 90/10 training and testing protocol, which I should say is not enough to demonstrate the accuracy of the method. As there are so little samples in the dataset (roughly 1500 samples, highly imbalanced), the model could be overfitted.

Furthermore, the authors did not used a third validation test, which should not be involved in the training nor used to stop the learning.

In order to make the experiments convincing, the authors should repeat the experiments with a proper protocol. For instance 70% for training, 15% for validation and 15% for testing.

Author Response

Our model was validated using holdout validation which doesn't need to break down the dataset in train/test/validation images. According to a study of Raschka [1], the idea behind this approach is to introduce new, unseen data after the model finished its training process, this method is very common for deep-learning techniques [2 - 4]. A standard split would be 2/3 for training and 1/3 for testing, but it is not uncommon to use our 90/10 split. 

Since our dataset was limited, another approach such as 70/15/15 training/test/validation would have taken too much data from our training set. 

The manuscript was updated to state this methodology. 

[1] Raschka, S., 2018. Model evaluation, model selection, and algorithm selection in machine learning. arXiv preprint arXiv:1811.12808.

[2] Meyer, G.P., Laddha, A., Kee, E., Vallespi-Gonzalez, C. and Wellington, C.K., 2019. Lasernet: An efficient probabilistic 3d object detector for autonomous driving. In Proceedings of the IEEE Conference on Computer Vision and Pattern Recognition (pp. 12677-12686).

[3] Fuchs, K., Grundmann, T. and Fleisch, E., 2019, October. Towards identification of packaged products via computer vision: Convolutional neural networks for object detection and image classification in retail environments. In Proceedings of the 9th International Conference on the Internet of Things (pp. 1-8).

[4] Meyer, G.P., Laddha, A., Kee, E., Vallespi-Gonzalez, C. and Wellington, C.K., 2019. Lasernet: An efficient probabilistic 3d object detector for autonomous driving. In Proceedings of the IEEE Conference on Computer Vision and Pattern Recognition (pp. 12677-12686).

Reviewer 2 Report

The revised version of the manuscript is significantly improved. The authors successfully replied to my comments/suggestions. I believe that the manuscript can be published and become a nice contribution.

Author Response

Thank you for all your feedback.